# Learning Robust Intervention Representations with Delta Embeddings

**Panagiotis Alimisis & Christos Diou**
Department of Informatics and Telematics
Harokopio University of Athens
Athens, Greece
{palimisis, cdiou}@hua.gr

## Abstract

Causal representation learning has attracted significant research interest during the past few years, as a means for improving model generalization and robustness. Causal representations of interventional image pairs (also called "actionable counterfactuals" in the literature), have the property that only variables corresponding to scene elements affected by the intervention / action are changed between the start state and the end state. While most work in this area has focused on identifying and representing the variables of the scene under a causal model, fewer efforts have focused on representations of the interventions themselves. In this work, we show that an effective strategy for improving out of distribution (OOD) robustness is to focus on the representation of actionable counterfactuals in the latent space. Specifically, we propose that an intervention can be represented by a Causal Delta Embedding that is invariant to the visual scene and sparse in terms of the causal variables it affects. Leveraging this insight, we propose a method for learning causal representations from image pairs, without any additional supervision. Experiments in the Causal Triplet challenge demonstrate that Causal Delta Embeddings are highly effective in OOD settings, significantly exceeding baseline performance in both synthetic and real-world benchmarks. Our code is publicly available.[1]

## 1 Introduction

Understanding how the world changes in response to actions and external interventions is fundamental for artificial intelligence agents, especially those operating in dynamic environments. Although deep learning models are highly successful at capturing complex patterns from data, they often fail to generalize to new situations where the underlying data distribution changes, which is a critical limitation for real world deployment Hendrycks et al. (2021); Geirhos et al. (2020). To overcome this, agents must recover the underlying mechanisms that generate and transform data, enabling causal reasoning and robust generalization (Pearl, 2009).

This fundamental problem falls within the scope of Causal Representation Learning (CRL) (Schölkopf et al., 2021), which seeks to disentangle the causal variables of a system (Khemakhem et al., 2020a). Despite its importance in practical applications such as robotics or healthcare (Gupta et al., 2024; Hellström, 2021; Sanchez et al., 2022; Tejada-Lapuerta et al., 2023), the challenge of learning disentangled and generalizable representations of the causal variables remains unresolved. Addressing this challenge requires accurate modelling of the underlying data generation process, a task which is guided by two fundamental assumptions within CRL. First, the Independent Causal Mechanisms (ICM) assumption, which posits that the distribution's generative process can be decomposed into autonomous and independent modules, each representing a distinct causal mechanism (Peters et al., 2017; Schölkopf et al., 2021). Second, the Sparse Mechanism Shift (SMS) assumption, which suggests that an intervention typically affects only a small, localized subset of

---

[1]Project page: `https://palimisis.github.io/Learning-Robust-Intervention-Representations-with-Delta-Embeddings/`

**Entangled Representations**

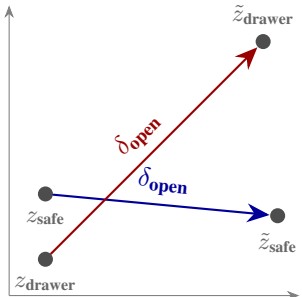

(a) Baseline Model (ERM). Action representations depend on the object and scene features.

**Consistent Representations**

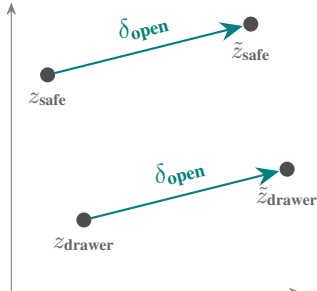

(b) Causal Delta Embeddings. The action representation $\delta_{open}$ is invariant to the object and scene context.

**Drawer: Open**                          **Safe: Open**

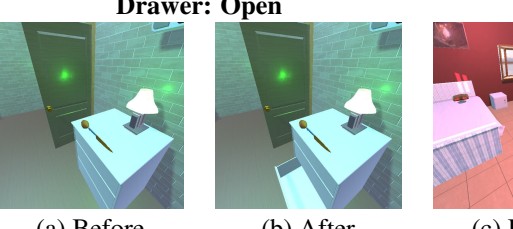

(a) Before    (b) After    (c) Before    (d) After

(c) Examples of intervention pairs from our dataset, showing pre- and post-intervention states for various actions and objects.

Figure 1: Visualizing Causal Delta Embeddings. Unlike a baseline model that produces entangled representations of action vectors (left), our model learns object invariant action representations (right), that generalize well to out of distribution samples. The model is trained on intervention pairs like those shown at the bottom.

these causal mechanisms (Schölkopf et al., 2021). While foundational methods focused on identifying these mechanisms from observational data or weak supervision (Brehmer et al., 2022; Lippe et al., 2022; Buchholz et al., 2023), recent advances have successfully leveraged interventional data to establish stronger identifiability guarantees, utilizing score-based methods (Kulkarni et al., 2025), unknown multi-node interventions (Varıcı et al., 2024), or invariance principles (Yao et al., 2024). However, the primary focus of these approaches remains on recovering the latent state variables. Fewer methods have focused on learning generalizable representations of actions (interventions), which can be equally important in predicting the outcome of interventions, especially when faced with novel situations.

In this paper, we introduce Causal Delta Embedding (CDE), a novel framework for learning robust representations of interventions from image pairs. Using CDEs the intervention can be effectively isolated and represented as the vector difference between the latent representations of pre- and post-intervention states if it satisfies the properties of (a) independence to causally irrelevant elements of the scene, in accordance to the ICM assumption (b) sparsity, in accordance to the SMS assumption and (c) object invariance, i.e., that the representation remains the same across objects. Using these properties as a guide, a learning strategy is proposed for learning CDEs from interventional image pairs.

We evaluate CDE on the Causal Triplet challenge (Liu et al., 2023), which encompasses 3 increasingly complex settings: single-object synthetic data, multi-object synthetic data and real world scenes from Epic Kitchens (Damen et al., 2022). Our experiments demonstrate that CDE establishes a new state of the art in OOD generalization for this challenge. Beyond quantitative performance, our analysis reveals that CDE learns semantically meaningful representations in the intervention space, autonomously discovering anti-parallel relationships between opposing actions (e.g., `open` vs. `close`) without any explicit supervision.

Our main contributions are as follows:

- We introduce *Causal Delta Embedding (CDE)*, a novel approach for learning generalizable representations of interventions in a disentangled latent space.
- We propose a multi-objective loss function, designed to learn well separated, sparse and object invariant causal representations directly from visual data.
- We perform an extensive quantitative evaluation showing that our approach achieves state-of-the-art results in the Causal Triplet challenge.
- We show that our model discovers the semantic structure of the intervention space, including fundamental anti-parallel relationships between opposing actions, without any explicit supervision.

## 2 RELATED WORK

**Causal Representation Learning**    Research on CRL spans multiple directions. One line of work focuses on identifying latent causal variables from high-dimensional observations (Khemakhem et al., 2020a). These methods established identifiability conditions for nonlinear ICA and demonstrated causal factor recovery under specific assumptions (Wendong et al., 2023; Monti et al., 2020; Khemakhem et al., 2020b). Recent theoretical advances leverage score-based methods to achieve identifiability even with unknown multi-node interventions (Varıcı et al., 2024; Varici et al., 2023), while practical applications demonstrate use cases in robotics (Kulkarni et al., 2025) and single-cell genomics (Lopez et al., 2023). Another body of work focuses on causal disentanglement (Yang et al., 2021; Shen et al., 2020; Locatello et al., 2020a), often extending the VAE framework (Kingma et al., 2013; Higgins et al., 2017) and leveraging interventional data (Brehmer et al., 2022; Lippe et al., 2022; Squires et al., 2023; Lippe et al., 2023; Ahuja et al., 2022; 2023; Buchholz et al., 2023) while other methods focus on object-centric representations in order to disentangle visual scenes into manipulable objects (Locatello et al., 2020b; Seitzer et al., 2022). The invariance principle has emerged as a unifying framework (Yao et al., 2024), showing that many CRL methods exploit distributional symmetries created by interventions rather than requiring explicit causal semantics. While most of these methods focus on recovering causal variables and their relationships, our work takes a fundamentally different approach: we learn embeddings that represent the interventions themselves, capturing how mechanisms change rather than identifying which variables exist.

**Visual Action Recognition and OOD Generalization**    Traditional action recognition methods rely on spatiotemporal patterns and achieve strong performance under IID conditions (Carreira & Zisserman, 2017; Feichtenhofer et al., 2019; Arnab et al., 2021). However, these correlation-based approaches struggle with distribution shifts (Geirhos et al., 2020) and often rely on spurious correlations (Wang & Jordan, 2024). Recent work has explored domain adaptation (Chen et al., 2019; Munro & Damen, 2020) and causal approaches (Magliacane et al., 2018; Wang et al., 2023) for robust action understanding. Another category of methods uses large Vision Language Action (VLA) models (Kim et al., 2024; Zitkovich et al., 2023; Ma et al., 2024) to enable agents to perform actions in challenging environments. These models typically depend on large-scale pre-training on diverse data, yet generalization to unseen tasks remains an open challenge (Sapkota et al., 2025). Unlike these approaches, our method learns *causal* representations of interventions that satisfy properties resulting from CRL assumptions, generalizing to novel object-action combinations without fine-tuning.

**Contrastive Learning and Sparse Representations**    Contrastive learning effectively learns meaningful representations by contrasting similar and dissimilar examples (Chen et al., 2020; Khosla et al., 2020; He et al., 2020; Grill et al., 2020). Recent theoretical work establishes deep connections between contrastive learning and causal structure discovery. Multi-view contrastive methods handle partial observability by learning shared representations across modalities (Yao et al., 2023; Federici et al., 2020; Tian et al., 2020). When data augmentations correspond to causal interventions, contrastive learning can provably disentangle causal factors (Von Kügelgen et al., 2021; Zimmermann et al., 2021). However, these methods contrast individual samples or augmented views, whereas our approach contrasts relationships between pre- and post-intervention pairs aiming at learning embeddings of the transformations themselves rather than sample level representations. The sparse

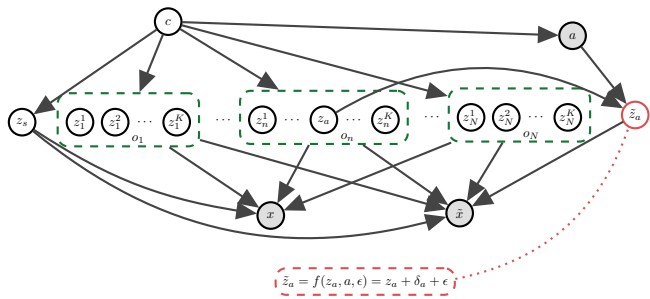

Figure 2: Causal Graph for a pair of observations $(x, \tilde{x})$ before and after an action $a$, proposed by Liu et al. (2023). The data generating process is described by a set of latent factors, including global scene level factors $z_s$ and local object level factors $z_n^k$, which are dependent due to confounders $c$. The action is assumed to influence only a few object level causal factors $z_a$ in the scene and the effect of that influence is captured by $\tilde{z}_a$. The red dashed line indicates the structural equation assumed by our CDE approach.

mechanism shift (SMS) principle (Schölkopf et al., 2021; Peters et al., 2017) suggests interventions affect only small subsets of causal mechanisms. This sparsity assumption enables identifiability in various settings, such as mechanism sparsity regularization for nonlinear ICA (Lachapelle et al., 2022; 2024) or instance dependent partial observability (Xu et al., 2024). Recent work demonstrates that combining sparsity with appropriate regularization improves both identifiability (Pfister & Peters, 2022; Buchholz et al., 2023) and generalization (Lopez et al., 2023; Layne et al., 2024). While adversarial training with sparsity constraints has been explored (Liu et al., 2023), it often yields poor OOD performance when confounders remain in the scene, motivating our stricter causal assumptions and explicit intervention embedding approach.

## 3 PROBLEM FORMULATION

The central challenge this paper addresses is the development of a CRL framework that can robustly infer actions / interventions from high-dimensional observations, particularly under distribution shifts.

We formalize this challenge within the framework of the Structural Causal Model presented by Liu et al. (2023) (Figure 2). Let us consider a set of causal variables $z \in \mathcal{Z} \subset \mathbb{R}^l$, representing the state of the underlying data generating mechanisms. These variables have dependencies that are defined through a set of structural equations. Assuming an additive noise model, these equations are of the form

$$z_i := f_i(\mathrm{pa}(z_i)) + \epsilon_i, \quad i = 1, \dots, l$$

where $\mathrm{pa}(z_i)$ denotes the set of causal parents of variable $z_i$, and the $\epsilon_i$ are mutually independent stochastic noise terms representing unmodeled exogenous factors. The high-dimensional visual observation $x \in \mathcal{X} \subset \mathbb{R}^d$ is rendered from these latent variables via a complex generative function $g : \mathcal{Z} \to \mathcal{X}$, such that $x = g(z)$.

We assume the latent variable $z$ can be partitioned into scene-level variables $z_s$ (e.g., illumination, camera pose) and a set of object-level variables $z_o = \{z_{n,k}\}_{n=1,k=1}^{N,K}$, corresponding to the $k$-th property of the $n$-th object. An agent performs an action $a \in \mathcal{A}$, which applies an intervention on the system. This intervention transforms the pre-intervention state $z$ into a post-intervention state $\tilde{z}$. Unobserved confounders ($c$) create spurious correlations and a training-testing distribution mismatch $P_{\text{train}}(Z = z, a) \neq P_{\text{test}}(Z = z, a)$. Following the *Independent Causal Mechanisms* principle (Schölkopf et al., 2012; Peters et al., 2017), we assume the true causal mechanism $P(\tilde{Z}_a = \tilde{z}_a | Z_a = z_a, a)$ is invariant to this shift. Therefore, a robust model must learn this invariant mechanism instead of non-stationary correlations.

We investigate two challenging types of OOD shifts:

- **Compositional Shifts:** Training and test sets share the same object classes, $O_{\text{train}} = O_{\text{test}}$, but disjoint sets of object-action pairs. $(A_{\text{train}} \times O_{\text{train}}) \cap (A_{\text{test}} \times O_{\text{test}}) = \emptyset$.

- **Systematic Shifts:** The training and test sets of object classes are disjoint, $O_{\text{train}} \cap O_{\text{test}} = \emptyset$.

Notice also that the exogenous noise $\epsilon$ may differ across states $z$ and $\tilde{z}$, especially in real-world data, where there is no control over the environment and data acquisition conditions. For this reason, they are referred to as "actionable counterfactuals" in Liu et al. (2023) (as opposed to perfect counterfactuals).

**Objective**   Given a dataset of paired observations $\mathcal{D} = \{(x, \tilde{x}, a)_j\}_{j=1}^M$, where $x$ and $\tilde{x}$ are the pre- and post-intervention images respectively, and $a$ is the corresponding action label, our objective is to learn a function $\mathcal{F} : \mathcal{X} \times \mathcal{X} \to \mathcal{A}$. This function must predict the action $a$ by learning a representation that isolates the invariant causal signature of the intervention, thereby achieving high performance on OOD test data characterized by the compositional and systematic shifts defined above.

## 4   CAUSAL DELTA EMBEDDINGS

Consider an *Encoder*, $\phi : \mathcal{X} \to \mathcal{Z}$ that maps a high-dimensional observation $x \in \mathcal{X}$ to a point in the latent space $\mathcal{Z}$. A Delta Embedding is defined as follows.

**Definition 1 (Delta Embedding)** *Given a pair of observations* $(x, \tilde{x})$ *corresponding to the state of the world before and after an intervention* $a \in \mathcal{A}$, *the Delta Embedding,* $\delta_a$, *is the vector difference*

$$\delta_a := \phi(\tilde{x}) - \phi(x)$$

Assuming identical noise across observations (i.e., perfect counterfactuals, the actionable counterfactual case will be discussed in the following section) and that $z_a$ in the data generating process of Figure 2 is identifiable by the encoder, for the Delta Embedding we have

$$\delta_a = [0 \quad \cdots \quad \tilde{z}_a - z_a \quad \cdots \quad 0]^T \tag{1}$$

where $z_a$ is the dimension (or subset of dimensions) of object $n$ that is affected by action $a$ in accordance with the notation used in the model of Figure 2. From equation 1 we observe the following properties of $\delta_a$.

1. *Independence*. Under the model of Figure 2, an action's representation is independent of scene properties and objects not affected by $a$, i.e., it is not influenced or informed by them.

2. *Sparsity*. If the assumption of *Sparse Mechanism Shifts* Schölkopf et al. (2021); Liu et al. (2023) holds, then the action $a$ will affect only a few underlying causal factors of the system, and the representation of the change, $\delta_a$, will be sparse.

To generalize to novel compositions of actions and objects, the action representation must satisfy additional properties. Specifically, even if the *Independence* and *Sparsity* properties are satisfied, if the action $a$ affects different objects in a different way, a learning system will not be able to predict how the action will modify the representations of unseen objects, or even seen objects but without any examples of these objects with $a$ in the training set.

We therefore introduce an additional requirement on the action representation, namely that it remains similar when applied to different objects, e.g., that the representation of action `open` is fundamentally the same, regardless of whether it is a door or a box that is being opened. We therefore introduce an additional property:

3. *Invariance*. The action representation $\delta_a$ should not vary across different objects. One way to formalize this is through the variance of the delta embeddings across samples, i.e.,

$$\text{Var}_{x \sim P(X)}[\delta_a(x)] \approx \mathbf{0} \tag{2}$$

**Definition 2 (Causal Delta Embedding)** *A Causal Delta Embedding (CDE) is a Delta Embedding that satisfies the properties of Independence, Sparsity and Invariance.*

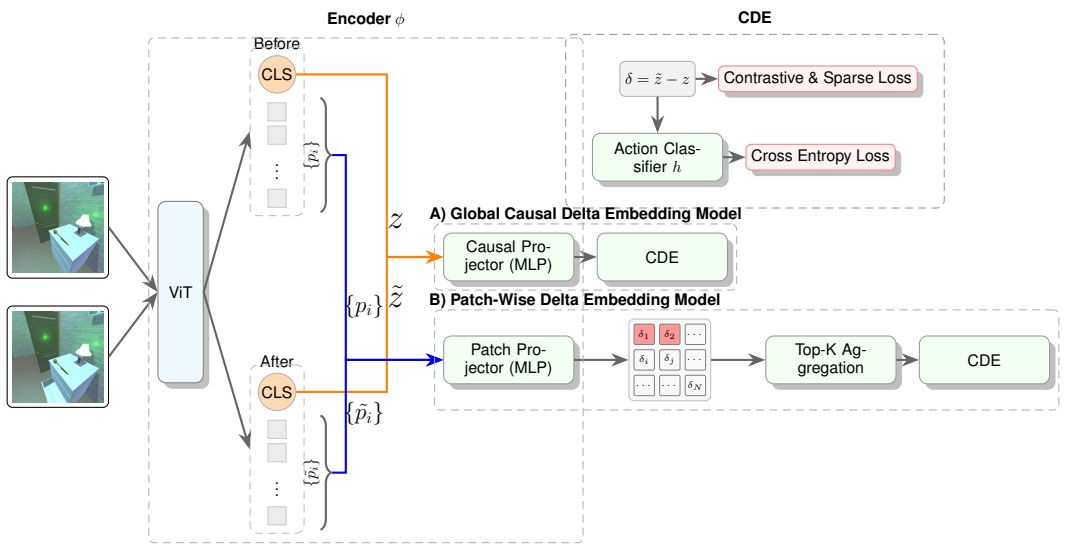

Figure 3: Model architecture. Model A (top) computes a global causal delta from CLS tokens. Model B (bottom) computes patch-wise deltas, aggregated to a causal delta. Both feed into a common action classifier.

In terms of the SCM of Figure 2, Causal Delta Embeddings can be implemented by defining the structural equation of $\tilde{z}_a$ as

$$\tilde{z}_a = f(z_a, a, \epsilon) = z_a + \delta_a + \epsilon \tag{3}$$

where $\epsilon$ is the value of zero mean, independent exogenous noise. When the only change before and after the intervention is the effect of action $a$ (perfect counterfactual), $\epsilon = 0$. The following section uses this definition to develop a strategy for learning Causal Delta Embeddings.

## 5 APPROACH

### 5.1 THE GLOBAL CAUSAL DELTA EMBEDDING MODEL

#### 5.1.1 MODEL ARCHITECTURE

We first introduce a *global* model, i.e., a model that learns a single causal representation from the entire image. The model consists of three main components, as illustrated in Figure 3 (A).

**The Encoder ($\phi$):** The encoder is responsible for mapping an input image $x$ into the $\mathcal{Z}$. It is composed of two sub-modules. (i) **A Pre-trained Vision Backbone:** We use a powerful Vision Transformer (ViT) (Dosovitskiy et al., 2020), specifically one pre-trained with the DINO self-supervision algorithm (Caron et al., 2021). The backbone processes the input image and outputs a high-dimensional feature vector. We use the output corresponding to the '[CLS]' token as the global image representation. (ii) **A Causal Projector:** The feature vector from the backbone is then passed through a small multi-layer perceptron (MLP). This projector's role is to transform the general-purpose features into an $l$-dimensional representation satisfying the Causal Delta Embedding properties.

**Delta Computation and the action classifier $h$:** We compute the CDE according to Definition 1. Given the latent vectors for the pre-intervention image ($z = \phi(x)$) and post-intervention image ($\tilde{z} = \phi(\tilde{x})$), the delta is calculated via simple, element-wise subtraction, $\delta = \tilde{z} - z$. This vector is the sole input to a final classification head, which is an MLP that takes the $l$-dimensional delta and outputs logits for the different action classes in $\mathcal{A}$.

### 5.1.2 IMPLEMENTATION OF THE LEARNING OBJECTIVE

To learn CDEs that satisfy the properties outlined in the Section 4, we combine three loss functions. (i) **Cross-Entropy Loss:** The primary objective is to ensure the delta embedding is useful for the downstream task. We use a standard Cross-Entropy loss, $\mathcal{L}_{\text{CE}}$ between the predicted action logits $h(\delta_i)$ and the one-hot ground-truth action label $a_i$. (ii) **Supervised Contrastive Loss:** To learn embeddings that are clustered together for the same action (Property 3, Invariance), we use the Supervised Contrastive Loss, $\mathcal{L}_{\text{contrast}}$ (Khosla et al., 2020). For a batch of $B$ delta embeddings, the loss for each embedding $\delta_i$ (the "anchor") encourages it to be closer to other embeddings of the same class ("positives") than to all other embeddings in the batch.

$$\mathcal{L}_{\text{contrast}} = \sum_{i=1}^{B} \frac{-1}{|P(i)|} \sum_{p \in P(i)} \log \frac{\exp(\text{sim}(\delta_i, \delta_p)/\tau)}{\sum_{j \neq i} \exp(\text{sim}(\delta_i, \delta_j)/\tau)} \tag{4}$$

where $P(i)$ is the set of all positive samples for anchor $i$ in the batch, $\text{sim}(\cdot, \cdot)$ denotes the cosine similarity, and $\tau$ is a scalar temperature hyperparameter. This loss component is also consistent with the structural equation 3. Finally, we introduce a (iii) **Sparsity Regularizer:** To encourage a minimal representation in line with the sparse mechanism shift hypothesis (Property 2, Sparsity), we apply an $\ell_1$ regularization penalty. This loss penalizes the sum of the absolute values of the embedding dimensions, promoting solutions where most dimensions are zero.

$$\mathcal{L}_{\text{sparsity}} = \frac{1}{B} \sum_{i=1}^{B} \|\delta_i\|_1 = \frac{1}{B} \sum_{i=1}^{B} \sum_{k=1}^{l} |\delta_{i,k}| \tag{5}$$

The final training objective is a weighted sum of these three components:

$$\mathcal{L}_{\text{total}} = \mathcal{L}_{\text{CE}} + \alpha_{\text{contrast}} \mathcal{L}_{\text{contrast}} + \alpha_{\text{sparsity}} \mathcal{L}_{\text{sparsity}} \tag{6}$$

where $\alpha_{\text{contrast}}$ and $\alpha_{\text{sparsity}}$ are scalar hyperparameters that balance the influence of each loss component. Note that the network is trained end-to-end, i.e., the encoder, $\phi$, is also updated during training.

### 5.1.3 ACTIONABLE COUNTERFACTUAL CASE

Notice that although no loss component explicitly enforces Property 1, this property is directly satisfied by the use of the Delta Embeddings and image pairs, where the observed changes are only due to $a$. This does not hold under the actionable counterfactual case (Liu et al., 2023), where the exogenous noise may change across observations, leading to $\epsilon \neq 0$ in equation 3 and non-zero elements in the delta embedding of equation 1. In the Appendix (Section E) we show that if $\epsilon$ is zero mean independent noise, then the classifier trained with $\mathcal{L}_{CE}$ is not affected by these non-zero values. This observation is also supported by our empirical findings presented in Section 6, where the delta embeddings lead to effectiveness improvements in real-world OOD data.

## 5.2 SPATIAL EXTENSION: THE PATCH-WISE MODEL

In complex scenes with multiple objects or significant background noise, an action may only affect a small, localized region of the image. A global embedding risks 'averaging out' this important local change, making it difficult to detect. To address this, we developed a patch-wise extension of our model.

### 5.2.1 ARCHITECTURE

The Patch-Wise model adapts the core architecture to operate on local regions, as shown in Figure 3 (B).

The architecture includes (i) **Patch-wise Feature Extraction:** We use a ViT backbone, but instead of taking the global '[CLS]' token, we retain the output feature vectors for each individual image patch. This gives us a sequence of patch features for both the before and after images, (ii) **Patch-wise Delta Computation:** A shared Causal Projector and the subtraction operation are applied independently to each corresponding pair of patch features. This yields a set of delta embeddings, $\{\delta_p\}$, one for

Table 1: Single-object ProcTHOR results. Our Global Delta Embedding model significantly improves OOD generalization under both compositional and systematic shifts. (R: ResNet-18, V: ViT-DINO)

| Method | IID Acc. | OOD Comp. | OOD Syst. | Gap Syst. ($\downarrow$) |
|--------|----------|-----------|-----------|----------|
| Vanilla-R | $0.96_{\pm 0.01}$ | $0.36_{\pm 0.13}$ | $0.48_{\pm 0.08}$ | 0.48 |
| Vanilla-V | $0.95_{\pm 0.01}$ | $0.34_{\pm 0.27}$ | $0.47_{\pm 0.11}$ | 0.48 |
| ICM-R | $0.95_{\pm 0.01}$ | $0.41_{\pm 0.15}$ | $0.50_{\pm 0.09}$ | 0.45 |
| ICM-V | $0.95_{\pm 0.01}$ | $0.38_{\pm 0.26}$ | $0.49_{\pm 0.01}$ | 0.46 |
| SMS-R | $0.96_{\pm 0.01}$ | $0.47_{\pm 0.18}$ | $0.54_{\pm 0.07}$ | 0.42 |
| SMS-V | $0.95_{\pm 0.01}$ | $0.34_{\pm 0.27}$ | $0.39_{\pm 0.04}$ | 0.56 |
| Ours(ViT-CLIP) | $\mathbf{0.97}_{\pm 0.01}$ | $0.91_{\pm 0.03}$ | $0.72_{\pm 0.02}$ | 0.25 |
| Ours(ViT-DINO) | $0.96_{\pm 0.01}$ | $0.91_{\pm 0.02}$ | $\mathbf{0.75}_{\pm 0.02}$ | $\mathbf{0.21}$ |
| Ours(ViT-MAE) | $0.96_{\pm 0.01}$ | $\mathbf{0.95}_{\pm 0.01}$ | $0.71_{\pm 0.02}$ | 0.25 |

each spatial patch location $p$. (iii) **Top-K selection:** We assume that the action's primary effect is localized to a small number of patches. We therefore identify the $k$ patches with the largest change by measuring the $L_2$ norm of their delta vectors ($\|\delta_p\|_2$). The loss $\mathcal{L}_{\text{total}}$ of equation 6 is then applied to each of the $k$ delta vectors, leading to one loss $\mathcal{L}_{\text{total}}^{(i)}$ for each vector. The final loss is then the average of the individual patch losses, $\mathcal{L} = \frac{1}{k} \sum_{i=1}^{k} \mathcal{L}_{\text{total}}^{(i)}$.

# 6 EXPERIMENTS

This section evaluates the effectiveness of our CDE framework. We first describe our experimental setup, then present the main quantitative results demonstrating CDE's effectiveness in OOD generalization, followed by qualitative and ablation analyses that provide deeper insights into its learned representations and design choices.

## 6.1 EXPERIMENTAL SETUP

Our evaluation is conducted on the Causal Triplet benchmark (Liu et al., 2023), specifically designed for intervention-centric causal representation learning. This benchmark features three distinct settings of increasing visual complexity: single-object synthetic scenes, multi-object synthetic scenes (both from ProcTHOR (Deitke et al., 2022)), and challenging real-world scenes from Epic-Kitchens (Damen et al., 2022). In all settings models are trained on pairs of pre- and post-intervention images with action labels and are evaluated for their ability to infer the action. Further details on the datasets and data filtering procedures are provided in the Appendix.

We follow the Causal Triplet protocol, evaluating models on both IID and OOD test sets. The OOD splits test two forms of generalization: *Compositional Distribution Shifts*, where the model encounters unseen combinations of actions and objects (e.g., open(drawer) when only open(door) and close(drawer) where seen during training); and *Systematic Distribution Shifts*, where generalization to entirely unseen object classes is required. Visualizations of these distribution shifts are available in the Appendix. All reported quantitative results are mean accuracies and standard deviations average over 3 random seeds. We set $\alpha_{\text{contrast}} = 2.0$ and $\alpha_{\text{sparsity}} = 1.0$ for all experiments (see the Appendix for more details).

We compare our two proposed models against the baselines from the Causal Triplet paper (Liu et al., 2023), including vanilla ResNets (He et al., 2016), methods incorporating causal regularization (ICM, SMS), and object-centric approaches (Slot Attention (Locatello et al., 2020b), GroupViT (Xu et al., 2022)).

## 6.2 MAIN QUANTITATIVE RESULTS

Our CDE framework consistently delivers substantial improvements in OOD accuracy across all evaluation settings, establishing a new state of the art. For single-object scenes, our global CDE model cuts the generalization gap from 0.56 to 0.21 while matching IID accuracy (Table 1). In

Table 2: Results across multi-object ProcTHOR and Epic-Kitchens (systematic shift). Oracle-mask utilizes a ground truth mask to isolate the intervened object. Conversely, other approaches must infer the action and object without this supervision.

| Dataset | Method | IID Acc. | OOD Acc. | Gap |
|---|---|---|---|---|
| ProcTHOR | ResNet | $0.83_{\pm 0.01}$ | $0.30_{\pm 0.08}$ | 0.53 |
| | Oracle-mask | $0.90_{\pm 0.01}$ | $0.42_{\pm 0.06}$ | 0.48 |
| | Slot-avg | $0.49_{\pm 0.01}$ | $0.15_{\pm 0.01}$ | 0.34 |
| | Slot-dense | $0.51_{\pm 0.01}$ | $0.19_{\pm 0.03}$ | **0.32** |
| | Slot-match | $0.66_{\pm 0.01}$ | $0.21_{\pm 0.01}$ | 0.45 |
| | Ours$_{(ViT-MAE)}$ | $0.91_{\pm 0.1}$ | $0.30_{\pm 0.02}$ | 0.61 |
| | Ours$_{(ViT-DINO)}$ | $0.92_{\pm 0.0}$ | $0.45_{\pm 0.03}$ | 0.47 |
| | Ours$_{(ViT-CLIP)}$ | $\mathbf{0.94}_{\pm 0.00}$ | $\mathbf{0.48}_{\pm 0.07}$ | 0.46 |
| Epic-Kitchens | ResNet | $0.42_{\pm 0.03}$ | $0.17_{\pm 0.03}$ | 0.25 |
| | CLIP | $0.45_{\pm 0.02}$ | $0.24_{\pm 0.02}$ | 0.21 |
| | Group-avg | $0.47_{\pm 0.03}$ | $0.24_{\pm 0.03}$ | 0.23 |
| | Group-dense | $0.50_{\pm 0.04}$ | $0.26_{\pm 0.03}$ | 0.24 |
| | Group-token | $0.52_{\pm 0.03}$ | $0.27_{\pm 0.03}$ | 0.25 |
| | Ours$_{(ViT-MAE)}$ | $0.50_{\pm 0.02}$ | $0.30_{\pm 0.02}$ | **0.20** |
| | Ours$_{(ViT-DINO)}$ | $0.54_{\pm 0.1}$ | $0.33_{\pm 0.00}$ | 0.21 |
| | Ours$_{(ViT-CLIP)}$ | $\mathbf{0.59}_{\pm 0.03}$ | $\mathbf{0.34}_{\pm 0.01}$ | 0.25 |

challenging multi-object and real-world settings (Table 2), our Patch-Wise model outperforms all baselines, including oracle methods that use ground-truth segmentation masks. Notice that the ResNet18 encoder performs significantly worse than the rest of the encoders, possibly indicating that this backbone cannot disentangle latent representations that depend on the actions from the rest of the scene as required by the model of Figure 2.

## 6.3 ACTION RELATIONSHIPS IN CAUSAL DELTA SPACE

To study the semantic structure of the learned delta space, we investigated whether the model could discover fundamental relationships between actions on its own. We computed the pairwise cosine similarity between all learned action representations. The result is visualized in the appendix (Figure 7). The analysis reveals that the model has learned a perfect *anti-parallel relationship* for opposite actions. The cosine similarity between the representations for open and close, for dirty and clean, as well as for turn on and turn off, is -1.0. This demonstrates that our framework not only separates the action concepts but also discovers opposing relationships between them, organizing the representations in a meaningful way. A similar pattern is observed in the more challenging real-world dataset where the model learns the anti-parallel representations for the open and close action pair, as well as for the fold and stretch pair (see Figure 8 in the Appendix for details).

In summary, the combination of strong predictive properties and consistent semantic structure demonstrates that our CDE framework learns meaningful representations of interventions. For further geometric analysis of the delta space, including UMAP projections, refer to the Appendix.

## 6.4 ABLATION STUDY

To understand the contribution of each component of our CDE framework, we also conducted a series of ablation studies, by analyzing the impact of each major loss component on the performance of our primary model with a ViT-DINO backbone. Table 3 presents the results, comparing our full model against versions where each loss component is removed, and a baseline trained only with standard CE loss.

The results demonstrate the effectiveness of our approach. Our full model achieves an OOD accuracy of 75.0%, a +8 point improvement over delta embedding representation trained solely with

Table 3: Ablation study of our method's components on the ViT-DINO model. Results are for the single-object systematic shift setting, showing the impact on OOD accuracy when each core component is removed.

| Model Configuration | IID Acc. (%) | OOD Acc. (%) |
|---|---|---|
| **Full Model** | **0.96** | **0.75** |
| *Ablations* | | |
| w/o Sparsity Loss | 0.96 | 0.73 |
| w/o Contrastive Loss | 0.95 | 0.68 |
| CE Loss only | 0.94 | 0.67 |
| Baseline (Liu et al., 2023) | 0.95 | 0.47 |

a CE objective, validating that explicitly structuring the representation space leads generalization improvements. Removing the supervised contrastive loss component causes a 7-point drop in OOD accuracy. Removing the sparsity loss term causes 2-point drop. Please refer to the Appendix for further ablation experiments.

# 7 CONCLUSION

This paper introduces the *Causal Delta Embedding (CDE)* framework, a simple yet effective approach to interventional causal representation learning. By explicitly modeling interventions as delta vectors in a structured latent space, CDE inherently satisfies the properties of independence, sparsity and invariance, leading to improved generalization. Our empirical validation on the Causal Triplet challenge demonstrates that CDE achieves state-of-the-art OOD generalization, outperforming prior methods across synthetic and real world datasets. Beyond quantitative gains, we show that CDE learns semantically meaningful representations without supervision, where opposing actions have anti-parallel representations. Despite the promising results, we acknowledge that limitations remain for real-world data, since both IID and OOD accuracies are still low for real world deployment, and also the use of universal delta embeddings for each action limits its ability to capture context-dependent visual transformations of actions. Future research directions includes the development of mechanisms to increase the robustness of the method in real-world scenarios involving increased noise or occlusions, extending the framework to video streams for modeling causal dynamics in temporal sequences, and investigating compositional properties of delta embeddings to enable multi-step interventions and generalization to novel action sequences.

## REPRODUCIBILITY STATEMENT

The previous sections have outlined the main building blocks of the proposed method, as well as the approach followed in the experiments, with the Appendix providing additional information and results. The code to reproduce the experiments is attached as supplementary material (without any identifiable information of authors) and will be made publicly available upon acceptance. Finally, all experiments were carried out by strictly following the Causal Triplet benchmark (Liu et al., 2023) evaluation protocols, which relies on the publicly available ProcTHOR (Deitke et al., 2022) and Epic-Kitchens (Damen et al., 2022) datasets.

## ACKNOWLEDGMENTS

The work leading to these results has been funded by the European Union under Grant Agreement No. 101057821 (project RELEVIUM). Views and opinions expressed are however those of the authors and do not necessarily reflect those of the European Union or the granting authority (HaDEA). Neither the European Union nor the granting authority can be held responsible for them.

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

## A  DATASET DETAILS

This section provides further details on the datasets used in our evaluation.

**ProcTHOR**  The ProcTHOR dataset Deitke et al. (2022) provides synthetic indoor scenes. For our single-object scenes, each scene contains one manipulated object, ensuring a clear focus on the intervention. In multi-object scenes, multiple objects are present, increasing the visual complexity of the scene, although only one object is again manipulated. We follow the dataset generation and filtering procedures as described in Liu et al. (2023) to ensure consistency with the Causal Triplet benchmark.

**Epic-Kitchens**  The Epic-Kitchens dataset Damen et al. (2022) comprises real world egocentric videos of diverse kitchen activities. From this, we extract pre- and post-intervention image pairs. Unlike synthetic environments, Epic-Kitchens introduces significant real world challenges such as camera motion, varying lighting conditions, occlusions and dynamic backgrounds, making the task of isolating interventions particularly challenging. To ensure dataset quality, a two-stage filtering process using Grounding DINO Liu et al. (2024) for zero-shot object detection is applied. For each extracted pair, the pipeline verifies that the target object appears clearly in both frames with a detection confidence above a set threshold $t = 0.45$. This automated filtering removes cases with poor object visibility or excessive motion blur.

### A.1  VISUALIZATIONS OF OOD SHIFTS

Figures 4, 5 and 6 visually illustrate the compositional and systematic distribution shifts utilized in the Causal Triplet benchmark.

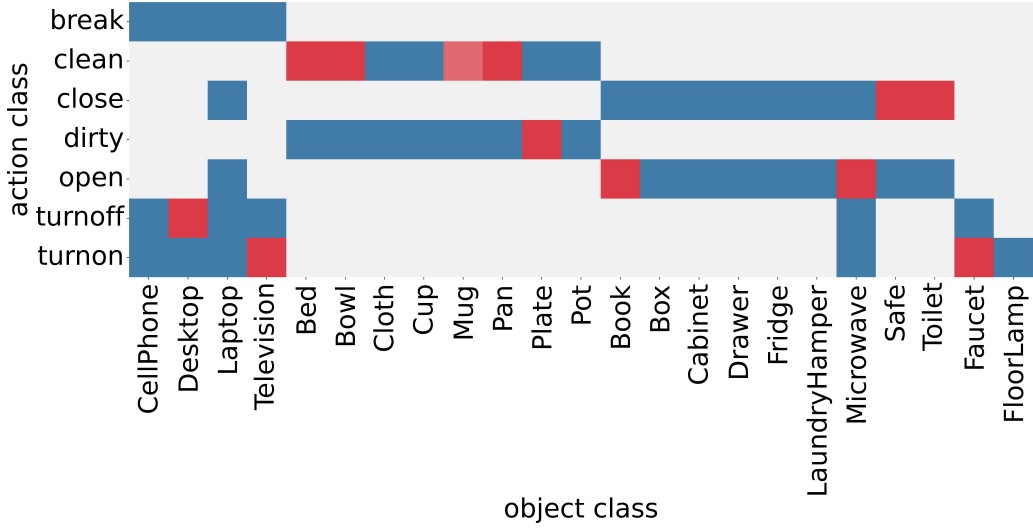

Figure 4: Compositional Distribution Shift in the ProcThor dataset. Blue boxes indicate IID data, while red boxes indicate novel OOD action-object combinations.

## B  GEOMETRIC ANALYSIS OF CAUSAL DELTA EMBEDDINGS

This section provides additional analysis of the geometric properties of the learned Causal Delta Embeddings, complementing the insights presented along with the experimental results.

### B.1  ACTION REPRESENTATION RELATIONSHIPS LEARNED FROM REAL-WORLD DATASETS

Figure 7 illustrates the the pairwise cosine similarities between the embeddings learned for all actions in the ProcTHOR dataset, while Figure 8 presents the same information for the more challeng-

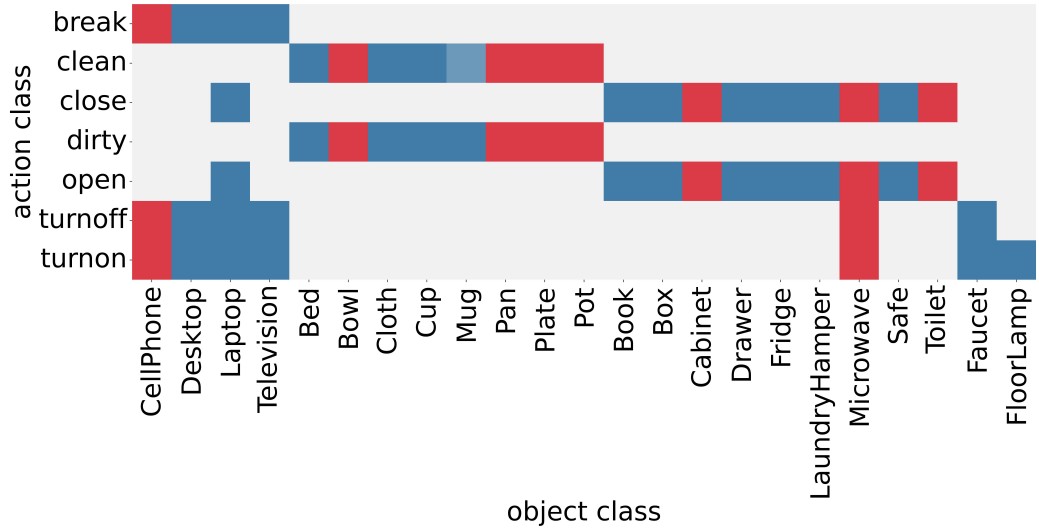

Figure 5: Systematic Distribution Shift in the ProcThor dataset. Blue boxes indicate IID data, while red boxes indicate novel OOD objects that the model has not encountered during training.

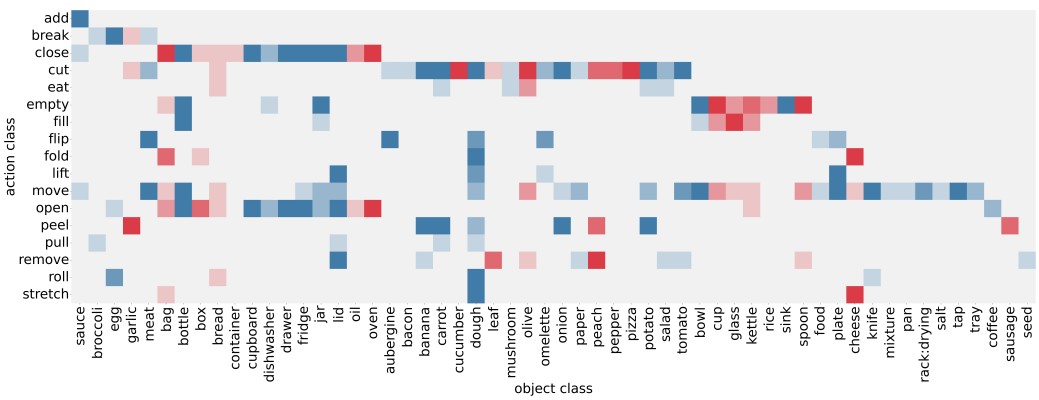

Figure 6: Systematic Distribution Shift in the EpicKitchens dataset. Blue boxes indicate IID data, while red boxes indicate novel OOD objects that the model has not encountered during training.

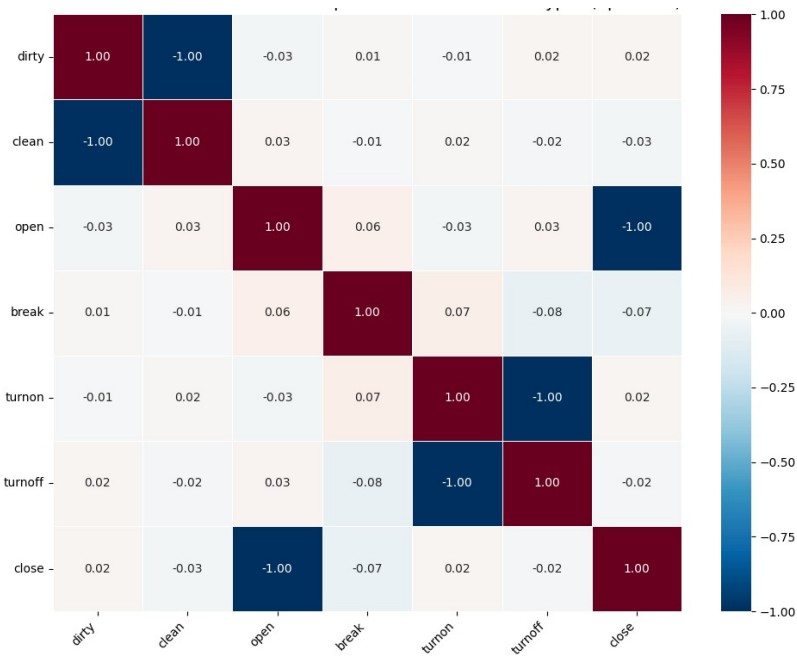

Figure 7: Heatmap of pairwise cosine similarities between all learned delta embeddings. The strong blue squares (similarity near -1.0) reveal a near-perfect anti-parallel relationship for opposite action pairs, which was discovered entirely from the data.

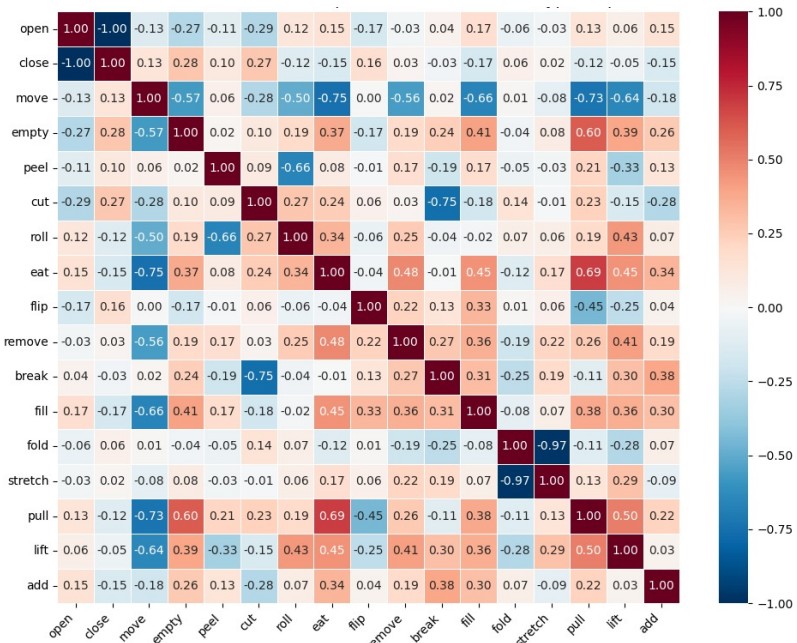

Figure 8: Heatmap of pairwise cosine similarities between all learned action prototypes for the EpicKitchens dataset.

ing real-world Epic Kitchens dataset. We observe that in both cases the learned relationships for certain opposing actions such as open and close as well as fold and stretch are antiparallel in the embedding space. Note, however, that the representation does not capture all antiparallel relationships (e.g., add and remove have a similarity of 0.19).

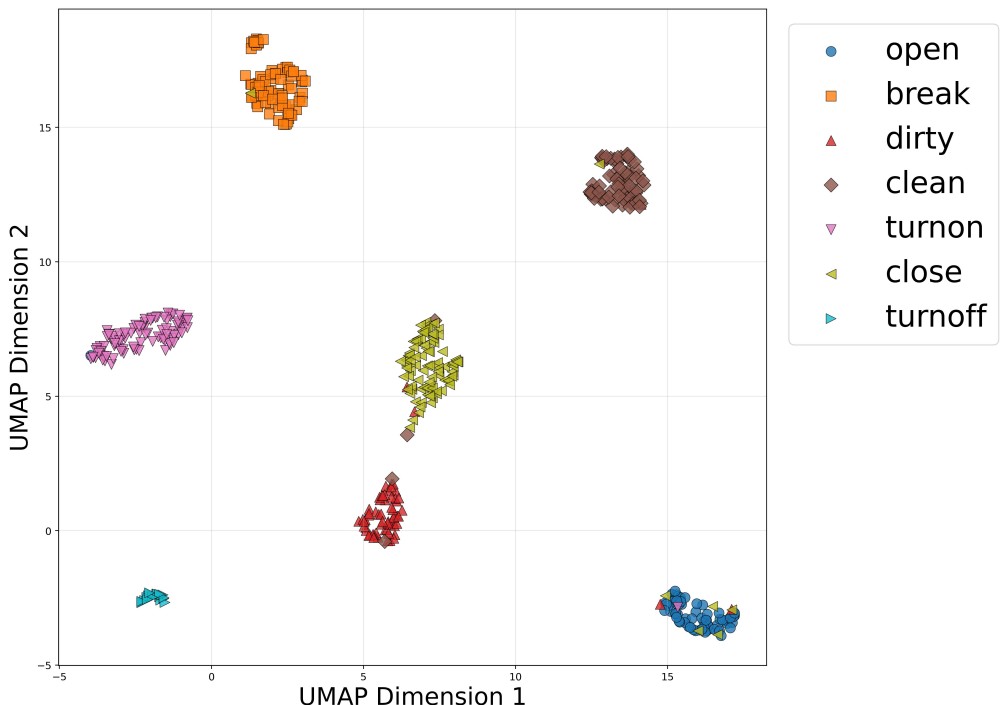

Figure 9: UMAP projection of individual delta embeddings from the IID test set. Embeddings are shaped by their ground-truth action. The plot reveals strong global separation between different action clusters.

## B.2 ANALYSIS OF LEARNED ACTION REPRESENTATIONS

To study the properties of the action representations resulting from our method, we first tested if the resulting delta embeddings could reliably predict the outcome of an intervention. To do this, for each sample in the OOD test set, we took the 'before' state embedding ($z$) and added the corresponding average action vector ($\mu_{action}$) that was computed using the training set samples. We then measured the cosine similarity between this predicted 'after' state and the ground-truth 'after' state ($\tilde{z}$). Our framework showed remarkable predictive power, achieving an average cosine similarity of $0.98$ in the single object systematic shift setting. This near perfect score confirms that the learned action prototypes function as true, generalizable transformation vectors, providing strong evidence that our model has learned the underlying mechanics of interventions.

## B.3 UMAP PROJECTION

Figures 9 and 10 present the UMAP projection of individual delta embeddings from the IID and OOD test set of the single-object environment respectively. The delta embeddings in the IID setting achieve a clear separation between each action, leading to a near perfect IID accuracy as was presented by our experiments. On the other hand, while strong intra-class cohesion is visible, the global separation between these action clusters is not always visually distinct in the OOD setting. This suggests that while representations remain locally coherent, action representations are not as clearly discriminated compared to the IID setting. It is worth mentioning, however, that the 2D projection may not fully capture the features of the high-dimensional latent space.

## C ABLATION STUDIES

In order to understand the effectiveness of each component of our method, we conducted a series of ablation studies to evaluate the impact of different backbone architectures, the impact of loss

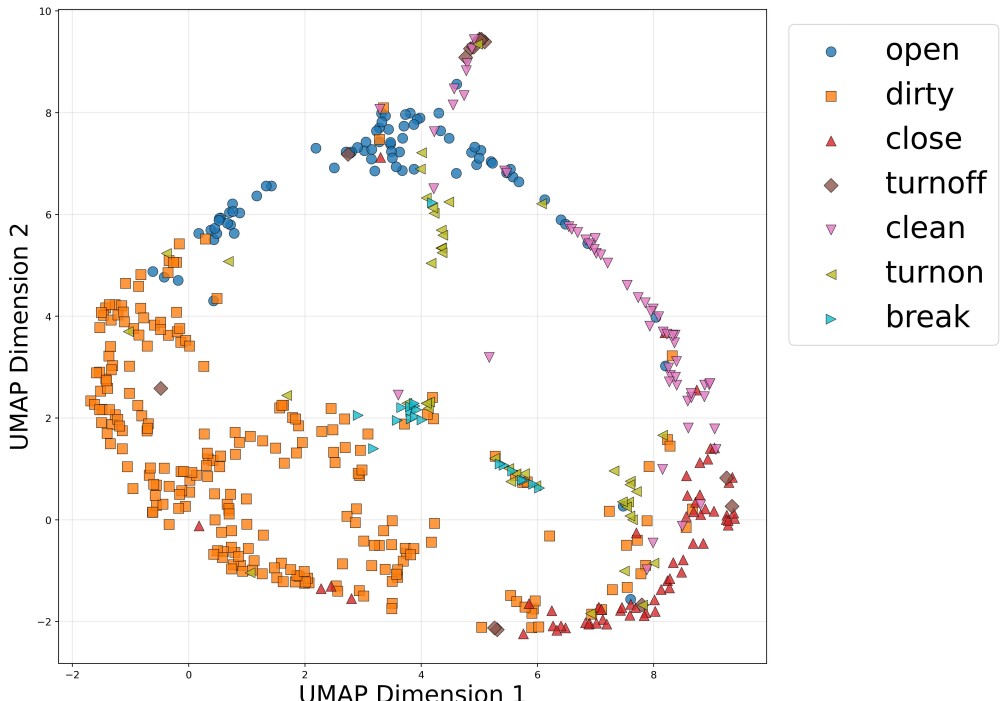

Figure 10: UMAP projection of individual delta embeddings from the OOD test set. Embeddings are shaped by their ground-truth action. The plot reveals strong local cohesion (points of the same shape cluster together) but shows a lack of clear global separation between the different action clusters.

hyperparameters $\alpha$ and the impact of the hyperparameter $k$ in the Top-K selection procedure for our Patch-Wise model. All the subsequent experiments ran on the single-object systematic shifts setting, except for the Top-K ablation study which ran on the multi-object systematic shifts setting. Also these experiments were performed in a smaller subset of the dataset.

Table 4 compares the OOD performance of the best configuration for each backbone against the benchmark's state-of-the-art ResNet-18 result. Our final ViT-based model significantly outperforms the best ResNet-based model, demonstrating that while the richer features from ViT enhance performance, the substantial gains are primarily driven by our proposed CDE learning framework.

Table 4: Comparison of OOD performance with different backbone architectures on the single-object systematic shift benchmark.

| Backbone | Method | OOD Acc. (%) |
|---|---|---|
| ResNet-18 | Liu et al. (2023) | 0.54 |
| ResNet-18 | Ours[*] | 0.45 |
| ViT-DINO | Ours (CE Only) | 0.67 |
| **ViT-DINO** | **Ours (Full Model)** | **0.75** |

[*] Best ResNet performance from our experiments was with CE + Con Loss.

## C.1 IMPACT OF BACKBONE ARCHITECTURE

To isolate the contribution of our CDE framework from the choice of feature extractor, we conducted a controlled comparison between our ViT-DINO backbone and the ResNet-18 backbone used in the original Causal Triplet benchmark.

## C.2 IMPACT OF LOSS HYPERPARAMETERS

In order to select values for $\alpha_{\text{contrast}}$ and $\alpha_{\text{sparsity}}$, we conducted an ablation study comparing various values and combinations between them. Table 5 compares some of the combinations of the values that we experimented with. Selecting a larger value for $\alpha_{\text{contrast}}$, rather than $\alpha_{\text{sparsity}}$, helps the model learn better representations, thus achieving better OOD accuracy. We set $\alpha_{\text{contrast}} = 2.0$ and $\alpha_{\text{sparsity}} = 1.0$ in all our main experiments.

Table 5: Comparison of various hyperparameter values for $\alpha_{\text{contrast}}$ and $\alpha_{\text{sparsity}}$ on the single-object systematic shift benchmark.

| $\alpha_{\textbf{contrast}}$ | $\alpha_{\textbf{sparsity}}$ | **OOD Acc. (%)** |
|---|---|---|
| 0.0 | 0.0 | $0.21_{\pm 0.02}$ |
| 0.1 | 1.0 | $0.27_{\pm 0.11}$ |
| 1.0 | 0.1 | $0.28_{\pm 0.04}$ |
| 0.5 | 0.5 | $0.29_{\pm 0.07}$ |
| 1.0 | 2.0 | $0.31_{\pm 0.07}$ |
| 2.0 | 1.0 | $\mathbf{0.33}_{\pm 0.07}$ |

## C.3 TOP-K SELECTION

In order to select the hyperparameter $k$ in multi-object and real world data settings, we executed an ablation study to understand the sensitivity of our method to this parameter. As presented in Table 6, we can see that OOD accuracy increases as $k$ increases too. This observation makes sense, since bigger objects (e.g. Fridge, Bed) would need more patches for their representations in order to be captured effectively. Thus, we set the value of $k = 4$ across all our multi-object and real world experiments.

Table 6: Comparison of OOD performance with $k$ values for the patch selection process in multi-object settings.

| **k** | **ProcTHOR** | **EpicKitchens** |
|---|---|---|
| $k = 1$ | $0.42_{\pm 0.07}$ | $0.12_{\pm 0.03}$ |
| $k = 2$ | $0.45_{\pm 0.06}$ | $0.13_{\pm 0.03}$ |
| $k = 3$ | $0.47_{\pm 0.04}$ | $0.13_{\pm 0.02}$ |
| $k = 4$ | $0.48_{\pm 0.04}$ | $0.15_{\pm 0.02}$ |

# D EXPERIMENTAL DETAILS

## D.1 HYPERPARAMETERS

Table 7 summarizes the key hyperparameters used across all experiments. These values were selected based on ablation studies and remained consistent across different experimental settings unless otherwise noted.

## D.2 EXECUTION ENVIRONMENT

All experiments were run on a NVIDIA A100 GPU with the Slurm Workload Manager. The code was implemented in Python, using the Pytorch library. Each run takes approximately one hour to complete for the ProcTHOR and two hours for the Epic-Kitchens dataset.

## D.3 IMAGE AUGMENTATIONS

We do not apply any augmentations to the images, since we do not want to modify the interventional nature of the pairs. Augmentation in this problem could harm our assumptions. For example, a rotation could affect equation 1 and eliminate the faithfulness of the encoder. We leave it as future work

Table 7: Summary of hyperparameters used across all experiments.

| Parameter | Value |
|---|---|
| Learning Rate | $1 \times 10^{-4}$ |
| Backbone LR | $1 \times 10^{-5}$ |
| Batch Size | 128 |
| Epochs | 50 (100 for Epic-Kitchens) |
| Weight Decay | 0.05 |
| $\alpha_{\text{contrast}}$ | 2.0 |
| $\alpha_{\text{sparsity}}$ | 1.0 |
| Temperature ($\tau$) | 0.07 |
| Top-K ($k$) | 4 |
| Embedding Dim. ($l$) | 256 (512 for Epic-Kitchens) |
| Input Resolution | $224 \times 224$ |

to investigate whether augmentations can boost OOD performance under different assumptions. We only resize images to $224 \times 224$ pixels and apply zero-mean normalization with unit variance.

### D.4 OPTIMIZATION

We use a batch size of 128 and an AdamW Loshchilov & Hutter (2017) optimizer with a cosine annealing learning scheduler for 50 epochs. In the real world setting, we instead train for 100 epochs. The ViT feature extractor is not frozen but fine-tuned with a reduced learning rate of $10\%$ of the network's base learning rate, which is set to $1 \times 10^{-4}$. The weight decay parameter is 0.05. All reported results include standard deviations computed over three independent runs with different random seeds.

### D.5 MODEL ARCHITECTURES

To ensure reproducibility and clarify the pre-training objectives used in our experiments, Table 8 details the specifications for the vision backbones used in our Causal Delta Embedding (CDE) framework and the comparative baselines. We utilize the implementations provided by the `timm` library.

Table 8: Comparison of Vision Backbones utilized in experiments.

| Model | Architecture | Pre-training Objective | Dataset | Params |
|---|---|---|---|---|
| **ViT-DINO** | ViT-S/16 | Self-Supervised (Distillation) | ImageNet-1k | 21.7M |
| **ViT-MAE** | ViT-B/16 | Self-Supervised (Reconstruction) | ImageNet-1k | 85.8M |
| **CLIP** | ViT-B/16 | Weakly-Supervised (Contrastive) | Web-400M | $\sim$86M |
| **ResNet-18** | ResNet-18 | Supervised (Classification) | ImageNet-1k | 11.7M |

## E ACTIONABLE COUNTERFACTUALS AND DELTA EMBEDDINGS

If we relax the requirement for identical noise across observations ($\epsilon \neq 0$ in equation 3), then the delta embedding of equation 1 becomes (assuming column vectors)

$$\delta_a^T = \begin{bmatrix} \epsilon_1, & \epsilon_2, & \ldots, & \tilde{z}_a - z_a + \epsilon, & \ldots, & \epsilon_l \end{bmatrix}^T$$

where $l$ is the number of vector dimensions and $\epsilon_1$, $\epsilon_2$ etc are assumed to be zero-mean noise variables that do not depend on $a$ or $\tilde{z}_a - z_a$. We use the following representation for convenience $\delta_a^T = \begin{bmatrix} \mathbf{u}_\epsilon^T, & \mathbf{u}_a^T \end{bmatrix}$, where $\mathbf{u}_\epsilon$ is the part of the representation that does not depend on $a$ and $\mathbf{u}_a$ the part that does. Consider that this representation is used with a binary logistic regression model with parameters $\mathbf{w}^T = \begin{bmatrix} \mathbf{w}_\epsilon^T, & \mathbf{w}_a^T \end{bmatrix}$. The gradient of the binary cross-entropy loss with respect to $\mathbf{w}_\epsilon$ is

$$\nabla_{\mathbf{w}_\epsilon} \left[ \mathcal{L}_{CE} \right] = \mathbb{E} \left[ \left( \sigma \left( \mathbf{w}_\epsilon^T \mathbf{u}_\epsilon + \mathbf{w}_a^T \mathbf{u}_a + b \right) - a \right) \mathbf{u}_\epsilon \right]$$

Note that at $\mathbf{w}_\epsilon = \mathbf{0}$ this becomes

$$\nabla_{\mathbf{w}_\epsilon} \left[\mathcal{L}_{CE}\right]|_{\mathbf{w}_\epsilon = \mathbf{0}} = \mathbb{E}\left[\left(\sigma\left(\mathbf{w}_a^T \mathbf{u}_a + b\right) - a\right) \mathbf{u}_\epsilon\right]$$

Given that $\mathbf{u}_\epsilon$ is independent of $a$, $\mathbf{u}_a$ and $b$ then this equation can be factorized as

$$\nabla_{\mathbf{w}_\epsilon} \left[\mathcal{L}_{CE}\right]|_{\mathbf{w}_\epsilon = \mathbf{0}} = \mathbb{E}_{\mathbf{u}_a, a}\left[\left(\sigma\left(\mathbf{w}_a^T \mathbf{u}_a + b\right) - a\right)\right] \mathbb{E}_{\mathbf{u}_\epsilon}\left[\mathbf{u}_\epsilon\right] = \mathbf{0}$$

since $\epsilon_1, \epsilon_2, \ldots, \epsilon_d$ are zero mean variables. This means that $\mathbf{0}$ is a stationary point of $\mathcal{L}_{CE}$. Since $\mathcal{L}_{CE}$ is convex for logistic regression, for any given $\mathbf{w}_a$ and $b$, $\mathbf{w}_\epsilon = \mathbf{0}$ is a global minimum of the loss. This implies that under imperfect interventions (actionable counterfactuals) under the zero mean independent noise assumption, the cross-entropy loss is minimized where the nonzero noise variables are ignored. Similar arguments apply in the multiclass case.

## F   ADAPTATION OF OBJECT-CENTRIC MODELS FOR ACTION PREDICTION

To leverage the latent structure of object-centric representations for downstream action reasoning, we adapt the implicit Slot Attention framework (Locatello et al., 2020b). The model decomposes each input scene into a set of $N$ spatially and semantically related regions (slots), each characterized by its own feature vector. Given a pair of pre- and post-intervention images, we explore three distinct aggregation strategies to bridge the object-centric slots with the action encoder.

- **Slot-Avg:** This baseline approach performs average-pooling over the $N$ slots for each image independently. The resulting single feature vectors for the pre- and post-intervention images are concatenated and passed to the action encoder, effectively treating the aggregated slots as a distributed representation.

- **Slot-Dense:** This strategy densely pairs every slot from the pre-intervention image with every slot from the post-intervention image, resulting in $N \times N$ combinations. All pairs are processed by the action encoder to generate relation embeddings, which are then aggregated via average-pooling to form the final action representation.

- **Slot-Match:** This method selectively pairs slots across the two images based on cosine similarity. Only the matched pairs are passed to the action encoder. The resulting $N$ relation embeddings are aggregated using max-pooling to capture the most significant latent changes corresponding to the intervention.

