# OpenReview forum: "Learning Robust Intervention Representations with Delta Embeddings"
_ICLR.cc/2026/Conference — ICLR 2026 Poster_

### Official Review · Reviewer_jtzb · 2025-10-30

**Soundness:** 3
**Presentation:** 3
**Contribution:** 3
**Rating:** 6
**Confidence:** 4

**Summary:**

The paper proposes Causal Delta Embeddings (CDE), a method to learn robust representations of interventions from image pairs. The key idea is to represent an intervention as the difference between pre- and post-intervention latent embeddings. The method combines cross-entropy, supervised contrastive, and sparsity losses to enforce independence, sparsity, and invariance. Results on the Causal Triplet benchmark show strong out-of-distribution generalization and meaningful representation structure.

**Strengths:**

- Clear motivation and solid theoretical grounding.
- Simple but elegant framework using delta embeddings to capture interventions.
- Strong experimental performance.
- Well-written and well-structured paper, with convincing ablations and visualizations.

**Weaknesses:**

- The method is trained with three losses. It is unclear how sensitive the performance is to the weighting between them. Although the appendix provides implementation details, it is not clear how one would set these weights in practice.

**Questions:**

See weaknesses.

---

> ### Author Response · Authors · 2025-11-19
>
> Thank you for your remarks. Following the review comments, we have uploaded a revised version of our paper which further clarifies the theoretical framework, provides experiments with additional backbones and explores the effect of actionable counterfactuals on our method. The ablation study of Table 3 indicates the contribution of each loss. Normally, one would select the weights using a hyperparameter tuning step. In the paper, we experimented in a subset of ProcTHOR using a small set of hyperparameters, and the results are discussed in Section C.2 and Table 5 of the Appendix. After that, these values are used throughout the experiments (including the experiments in EpicKitchens). One should therefore start with these values and, if possible, perform additional hyperparameter tuning before applying the method in a new dataset. So at this point the investigation of the effect of hyperparameters remains empirical, while future work can include a deeper investigation on the effects of each loss component.

---

### Official Review · Reviewer_AZZ2 · 2025-10-31

**Soundness:** 3
**Presentation:** 3
**Contribution:** 3
**Rating:** 4
**Confidence:** 3

**Summary:**

This paper proposes to learn representations corresponding to interventions, in contrast to learning representations for the underlying variables. The motivation behind this is to learn reusable representations/embeddings for interventions invariant of the object on which the intervention acts. From the paper, I gather that such representations are useful in Visual Language Action (VLA) models.

**Strengths:**

The idea of learning representations for actions is interesting, and it is a reasonable choice to learn these representations using interventional data. I believe Causal Delta Embeddings (CDEs) will have practical applications in robotics and other interactive domains.

**Weaknesses:**

I include my major concerns under "weaknesses" and minor concerns (mainly related to writing) under "questions." My most important concerns are W1, W2, and W5 (d, e). Most weaknesses/questions can be answered without experiments. I will raise my score if my concerns are addressed.

**W1. Interventional vs counterfactual**: A major confusion I have is whether CDE requires interventional or counterfactual data, the latter being a stricter requirement. In lines 194-197, the goal is stated to learn some function using a dataset of pre- and post-interventional data. If these data samples are obtained with the same exogenous noise, they are counterfactual samples. The equation in lines 157-158 mentions the exogenous noise $\epsilon$, but its role in the training dataset is unclear. "Identical noise" is assumed in lines 211, which implies the requirement is counterfactual data. To satisfy eq. (1) using interventional data, the encoder $\phi$ must disregard the exogenous noise completely. Can the authors clarify what the real requirements are?

**W2. Should the underlying representation be identifiable?** Even if counterfactual data is provided, eq. (1) can be satisfied only using an identifiable $\phi$. Is my understanding correct? If that's the case, will such an identifiable encoder $\phi$ be automatically learned while learning CDE, or is it a starting requirement? If my understanding is wrong, please provide a counterexample and include it in the text. I have a related concern on line 216. It is written "an action's representation is independent of the causally irrelevant..." Does "independent" here mean statistically independent, or that it is not affected by the "causally irrelevant elements?"

**W3. Will the aggregation module allow patch-level CDE?** In Sec. 5.2.1, the loss function is applied only over the aggregated embeddings. It is possible for patch-level embeddings to not follow Def. 2 even when the aggregated embedding satisfies Def. 2. Is that true? Are patch-level CDEs not required to follow Def. 2?

**W4. Comparison to BISCUIT**: BISCUIT (Lippe et al., 2023) also encodes the action variable responsible for interventions. See Fig. 7 in (Lippe et al., 2023). Is it possible to compare CDE against BISCUIT?

**W5. Questions on experiments**:

**W5. (a)** What are the baseline models trained on? Lines 257-259 say that the encoder is a DINO-pretrained vision backbone. Are ResNets and ViT in Tables 1 and 2 also DINO-trained? Why is it compared against a CLIP in Table 2, instead of another DINO? What exactly is the oracle-mask approach?

**W5. (b)** How are object-centric models, such as Slot Attention, adapted to predict actions?

**W5. (c)** Why is OOD Comp. accuracy smaller than OOD Syst. accuracy for all baselines, except CDE in Table 1? OOD Syst. seems to be a more difficult task.

**W5. (d)** How is CDE able to achieve that much accuracy in OOD Syst. in Table 1? The manifestations of actions in images are linked to the object on which it acts. So how can the model foresee what the action will look like on an unseen object? I can think of a possibility: CDE works only in scenarios where the action manifestation on the unseen object was seen during training, and that can also maybe explain the difference between OOD Comp. and OOD Syst. accuracies. Can the authors explain why CDE works on OOD Syst.?

**W5. (e)** The CE-only model in Table 3 beats all baselines in Table 1. I thought vanilla-R and vanilla-V were also trained with just CE. So does CDE benefit from something beyond its loss functions?

**W5. (f)** What is Fig. 10 supposed to convey? If the point was to show that CDEs are suitable for k-NN, then it must be compared with other baselines or other variants of CDE with fewer losses (like in Table 3). Although I would appreciate answering this question with experiments, I will not reduce my score if the experiments are not provided, as it is not a main experiment.

**Questions:**

**Q1. Related works on CRL from interventional data**: In related works, some works on causal representation learning (CRL) that use interventional data are mentioned. However, these works are not mentioned in the introduction when the story is built around the existing state of CRL. I think it is important to highlight the current advances in CRL using interventional data, and how this work is different from them. Also, I would suggest adding more recent works in CRL using interventional data. See [A1-3] and the references therein. [A1] is a contemporary work. The authors need not include it, but I shared it here as a source for recent works.

**Q2. Related works on contrastive learning**: There are two sentences on contrastive learning -- one listing two general contrastive learning works, and another providing a slightly unfair comparison w.r.t. this work. While contrastive learning only compares individual samples, it also does not require any pre- and post-intervention sample pairs, like this work. Another thing is that several works that link contrastive learning to CRL are missing. Some of them are [A4-5] and related works in [A6].

**Q3. Related works on SMS**: Again, a few important works on SMS are missing from your literature survey. See [A7-9].

**Q4. Minor writing comments**:

**Q4. (a)** I suggest rewording the sentence in line 121, starting with "While previous methods..." to clarify what part of the interventional mechanism is captured by CDEs and what invariance (across contexts) is targeted by CDE.

**Q4. (b)** Line 154: "a set of causal variables $Z\in\mathcal{Z}\subset\mathbb{R}^l$..." Which variable is the set here?

**Q4. (c)** Line 161: "a complex, **non-invertible** generative function..." How can you learn actions if the variables on which the action works are not retrievable?

**Q4. (d)** How did eq. (3) come about? Is $f$ in eq. (3) same as $f$ in line 157? How did the second equality in eq. (3) come about? Are the non-zero indices of $\delta_a$ aligned with the changes in $z_a$ due to the action?

**Q4. (e)** Can you clarify what the sentence in line 323 starting with "If, however, these..." means?

**Q4. (f)** I find lines 425-427 to be slightly misleading. In Fig. 8, there are indeed some anti-parallel representations learned. But there are also some nearly-anti-parallel representations for action pairs that do not make sense. For example, cut-break, move-pull and move-eat have around -0.75 in Fig. 8.

**Q4. (g)** Do any of the numbers in Tables 5 and 6 appear anywhere in the main results?

**References**

[A1] Pranamya Kulkarni, Puranjay Datta, Burak Varıcı, Emre Acartürk, Karthikeyan Shanmugam, Ali Tajer, "ROPES: Robotic Pose Estimation via Score-Based Causal Representation Learning", ArXiv 2025.

[A2] Burak Varıcı, Emre Acartürk, Karthikeyan Shanmugam, Ali Tajer, "Linear Causal Representation Learning from Unknown Multi-node Interventions", NeurIPS 2024.

[A3] Dingling Yao, Dario Rancati, Riccardo Cadei, Marco Fumero, Francesco Locatello, "Unifying Causal Representation Learning with the Invariance Principle", ICLR 2025.

[A4] Julius von Kügelgen, Yash Sharma, Luigi Gresele, Wieland Brendel, Bernhard Schölkopf, Michel Besserve, Francesco Locatello, "Self-Supervised Learning with Data Augmentations Provably Isolates Content from Style", NeurIPS 2021.

[A5] Roland S. Zimmermann, Yash Sharma, Steffen Schneider, Matthias Bethge, Wieland Brendel, "Contrastive Learning Inverts the Data Generating Process", ICML 2021.

[A6] Dingling Yao, Danru Xu, Sébastien Lachapelle, Sara Magliacane, Perouz Taslakian, Georg Martius, Julius von Kügelgen, Francesco Locatello, "Multi-View Causal Representation Learning with Partial Observability", ICLR 2024.

[A7] Elliot Layne, Jason Hartford, Sébastien Lachapelle, Mathieu Blanchette, Dhanya Sridhar, "Sparsity regularization via tree-structured environments for disentangled representations", ArXiv 2024.

[A8] Sébastien Lachapelle, Pau Rodríguez López, Yash Sharma, Katie Everett, Rémi Le Priol, Alexandre Lacoste, Simon Lacoste-Julien, "Nonparametric partial disentanglement via mechanism sparsity: Sparse actions, interventions and sparse temporal dependencies", ArXiv 2024

[A9] Danru Xu, Dingling Yao, Sébastien Lachapelle, Perouz Taslakian, Julius von Kügelgen, Francesco Locatello, Sara Magliacane, "A Sparsity Principle for Partially Observable Causal Representation Learning", ICML 2024.

---

> ### Author Response · Authors · 2025-11-19
>
> Thank you for your very careful review of our manuscript and the very insightful suggestions. We have tried to address all of them and the paper has been significantly improved as a result. We have uploaded a revised version of the paper, including additional experiments and theoretical investigations attempting to address your comments. Please find detailed answers to your concerns in the following.
>
> **"W1"**
> Thank you, this is a very good point and we understand why the formulation was confusing. We have revised the paper (Sections 3 and 4) to explicitly distinguish between the perfect counterfactual case (assumed in Eq. (1)) and the "actionable counterfactual" case, as denoted in Liu et al. 2023, where additional scene elements may change across a pair of observations (modeled via nonzero additive noise in our case). Moreover, we have added a brief proof in the appendix (Section E in the updated version of the paper), showing that if this noise is independent and has zero mean, then training with the CE loss (as done in our architecture, presented in Fig. 3) will lead the model to ignore these non-zero elements, supporting the use of the CDE approach even in this case. This also explains the performance improvements attained by the proposed approach in the EpicKitchens dataset, where indeed the perfect counterfactual assumption does not hold.
>
> **"W2"**
> The encoder should be able to disentangle between the part of the underlying representation that depends on $a$ and the rest. Specifically, in order for Eq (1) to hold, for every action $a$ the encoder must be able to identify $z_{a}$ (the part of the representation affected by $a$) and the rest of the representation must not be affected by $a$. For example, if an action affects the color of an object, then we want to have a part of $\phi{(x)}$ that depends only on the color and the rest of the representation to be independent of the color. In that sense we do not require full identifiability (e.g., to be able to disentangle shape features, if these are not affected by an action in our set of actions). In our experiments, we do not require that the encoder can identify $z_{a}$ a priori, since it is also updated during training. The ability of the trained encoder to identify $z_{a}$ is evaluated empirically through our experiments and it is highly likely that this is the reason for the observed difference in performance between the ResNet and the rest of the backbones (ResNet performs much worse, possibly due to its inability to identify $z_a$). We have added clarifications to highlight these points in Sections 4 and 6 of the updated manuscript.
>
> **"W3"**
> This is a very good point. We have updated the description of the method to calculate the loss function independently for each patch and then use the average of these losses as the total loss. We have also re-ran the experiments under this setting. The results are similar to the ones in the original paper (possibly due to the top-k filtering step).
>
> **"W4"**
> This comparison is possible but would require the we adapt the BISCUIT method to the Causal Triplet setting (it's not directly comparable). This is different from a comparison against an established baseline. We therefore chose not to compare against BISCUIT.

---

> > ### Author Response · Authors · 2025-11-19
> >
> > **"W5.a"**
> > **"What are the baseline models trained on? Lines 257-259 say that the encoder is a DINO-pretrained vision backbone. Are ResNets and ViT in Tables 1 and 2 also DINO-trained?"**
> > The ResNet model was pretrained in ImageNet1K for classification and ViT was DINO-trained in ImageNet1K. For clarification, we have added Appendix D.5 which details the specifications for the backbones used across all experiments in our paper. Moreover, we have added additional backbones, namely CLIP and MAE.
> >
> > **"Why is it compared against a CLIP in Table 2, instead of another DINO?"**
> > We had originally included CLIP since the paper of Liu et al. 2023 included results with a CLIP backbone. To avoid confusion and to enhance our experiments we now explicitly indicate the backbone used in each experiment and have added results with additional backbones.
> >
> > **"What exactly is the oracle-mask approach?"**
> > The Oracle-Mask method provides a ground truth mask to the intervened object, so it provides almost full information for the intervened object. Details are provided in Liu et al. 2023. We have added a clarification in the caption of Table 2. Based on the results, our method using 16x16 patches in a regular grid and top-k selection can effectively isolate the intervened object and classify the action without ground truth mask supervision.
> >
> > **"W5.b"**
> > Thank you, this was indeed not clearly specified in the original paper since this is based on the Causal Triplet paper. We have added Appendix F to explain this in detail. Given the pre- and post-intervention images, an aggregation strategy is applied (one of Slot-Avg, Slot-Dense and Slot-Match) to derive a global representation for the action encode from the object-centric slots.
> >
> > **"W5.c"**
> > All results with the ResNet backbone have been copied by the paper of Liu et al. 2023. The results using the ViT-DINO encoder have been performed by us. In all cases for the baseline, OOD Comp indeed performs worse than OOD Syst. We do not know why this happens. As you indicate, OOD Syst. is indeed a more difficult task, and this is depicted in the CDE results.
> >
> > **"W5.d"**
> > Qualitatively (no formal proof is available), the action representation is learned using the same action across different objects in the training set. Moreover, it is "forced" by the invariance property (implemented via the contrastive loss) to be the same across these different objects. The assumption is that if the properties of independence, sparsity and, most importantly in this case, invariance hold, then the action representation for unseen objects will remain similar.
> >
> > **"W5.e"**
> > The CE-only baseline in Table 3 refers to the delta embedding, while Vanilla-R is trained using the Causal Triplet approach. We have now clearly specified this and have added the causal triplet (Vanilla-V) as baseline in Table 3.
> >
> >
> > **"W5.f"**
> > We agree that this figure did not provide a clear message supporting the proposed approach. We therefore removed it in this updated version.

---

> > > ### Author Response · Authors · 2025-11-19
> > >
> > > **"Q1,Q2,Q3,Q4.a"**
> > > We have modified the related work section to include the suggested papers and make all appropriate changes.
> > >
> > > **"Q4.b"**
> > > We have updated the notation to use lowercase letters for consistency, so in $z \in \mathcal{Z} \subset \mathbb{R}^l$, $z$ is the variable from a set $\mathcal{Z}$ of $l$-dimensional vectors.
> > >
> > > **"Q4.c"**
> > > As indicated in response to W2, the generative function $g: \mathcal{Z} \to \mathcal{X}$ does not need to be invertible  (meaning we cannot perfectly recover the full latent state $Z$) since we do not attempt to invert $g$ to find $z$. Instead, we learn a specific encoder $\phi: \mathcal{X} \to \mathcal{Z}_{learned}$ guided by the action supervision. The key is that while the absolute state may not be fully recoverable, the change induced by the action produces a detectable signature in the pixel space. Our framework learns an encoder $\phi$ such that the difference vector $\delta = \phi(\tilde{x}) - \phi(x)$ isolates this  signature. Effectively, we only need to invert the interventional mechanism (the change), not the entire generative mechanism (the scene). The cross-entropy loss on the delta vector ensures that $\phi$ learns to extract exactly those features necessary to distinguish the action, even if the complete underlying state $Z$ remains partially unidentifiable. To avoid confusion (i.e., to avoid leading the reader to believe that we actually require the function to be non-invertible), we have now removed this word.
> > >
> > > **"Q4.d"**
> > > **"How did eq. (3) come about?"**
> > > Indeed, the way the equation was presented did not come naturally from the text. We have now explicitly identified that we assume an additive noise model and we have replaced the $\mathcal{N}$ noise with the more general $\epsilon$ exogenous independent noise, which is later assumed to be zero mean (to support the arguments related to the actionable counterfactual case, new Section 5.1.3 and Appendix E).
> > >
> > > **"Is $f$ in eq. (3) same as $f$  in line 157?"**
> > > Thank you for pointing this out. It is, in the updated version of the paper the theoretical model and equations as well as the relevant assumptions  are presented more clearly.
> > >
> > > **"Are the non-zero indices of $\delta_\alpha$ aligned with the changes in $z_\alpha$  due to the action?"**
> > > Yes, although we understand that this is not perfectly accurate notation, it is compatible to the notation used in the model of Figure 2, and we therefore hope it does not lead to confusion. We have added a sentence to make this explicitly clear.
> > >
> > > **"Q4.e"**
> > > We have expanded on this point, by adding Section 5.1.3 and a brief proof that this holds in Appendix E.
> > >
> > > **"Q4.f"**
> > > We acknowledge that Figure 8 shows some relationships with negative similarity ($-0.75$) that are not semantically opposite (e.g., cut-break, move-pull, move-eat). However these are not anti-parallel in the representation space. The anti-parallel relationships (similarity $\approx -1.0$) are all semantically opposite. There are also relationships that are semantically opposite (e.g., add and remove) that are not anti-parallel in the representation space (similarity $0.19$). In other words, the learned representation do not completely recover all the semantic relationships. Still, we find it interesting and worth mentioning that some of them are retrieved. We have added this clarification in the text regarding Figure 8.
> > >
> > > **"Q4.g"**
> > > These results were obtained using a subset of the data to assess the impact of hyperparameters, and this is the reason why they do not appear in the main text. This is now clearly specified in the text.
> > >
> > > We hope that the revised version of the paper as well as the responses to the reviewer comment have addressed the reviewer's concerns.

---

> > > > ### Comment · Reviewer_AZZ2 · 2025-11-26
> > > > **Response from reviewer AZZ2**
> > > >
> > > > I thank the authors for their response. Their rebuttal and revision have adequately addressed most of my concerns. Therefore, I increase my rating to reflect these changes.
> > > >
> > > > > **W1. Does CDE require counterfactual data?**
> > > >
> > > > This question is answered. The proof is App. E is sufficient. I think there is a small typo in the equation in line 1107. The gradient is written w.r.t. w_n instead of w_eps.
> > > >
> > > > > **W2. Should the underlying representation be identifiable?**
> > > >
> > > > It seems to me that partial identifiability is assumed in the theoretical definitions, but is not required in the experiments. This point must be made when CDE is introduced. Also, please see my question about the word "independent" in the original review.
> > > >
> > > > > **W3. Will the aggregation module allow patch-level CDE?**
> > > >
> > > > The authors' response is sufficient.
> > > >
> > > > > **W4. Comparison to BISCUIT**
> > > >
> > > > Can you provide more concrete reasons for not comparing? I am not insisting on comparing.
> > > >
> > > > > **W5. Questions on experiments (a-f)**:
> > > >
> > > > My questions were answered adequately. But overall, I feel all baselines were directly pulled from the Causal Triplets paper. I would recommend having comparison baselines tailored to show the utility of the proposed approach. An example is given below:
> > > >
> > > > From the Causal Triplets paper, vanilla-R and vanilla-V concatenate features from pre- and post-action images and then train action classifiers over them. Computationally, CDE seems to be a special case of vanilla-R and vanilla-V, where the input is explicitly the difference between features of pre- and post-action images. Can the authors explain why CE-only in Table 3 outperforms vanilla-R and vanilla-V on OOD accuracy (Table 1)?
> > > >
> > > > Additionally, that k-NN experiment might have conveyed a clearer picture of CDE, even if the results did not fully support CDE. I think that clarity is more important than just a method that gives better numbers on a benchmark.
> > > >
> > > > > **Q1-5. Writing concerns**
> > > >
> > > > Writing concerns are addressed.
> > > >
> > > > **Q4. (c)** This formulation of partial invertibility required for CDE must be exactly detailed under the assumptions.

---

> > > > > ### Author Response · Authors · 2025-12-01
> > > > >
> > > > > **"W1. Does CDE require counterfactual data?
> > > > > This question is answered. The proof is App. E is sufficient. I think there is a small typo in the equation in line 1107. The gradient is written w.r.t. $w_n$ instead of $w_{eps}$."**
> > > > >
> > > > > The typo has been fixed.
> > > > >
> > > > > **"W2. Should the underlying representation be identifiable?
> > > > > It seems to me that partial identifiability is assumed in the theoretical definitions, but is not required in the experiments. This point must be made when CDE is introduced. Also, please see my question about the word "independent" in the original review."**
> > > > >
> > > > > Thank you for this comment. We have updated the text and clarified in the introduction of the theoretical framework that only $z_a$ must be identifiable. We also clearly indicate that the encoder $\phi$ is updated during training. Regarding independence, we clarify that "Under the model of Figure 2, an action's representation is independent of scene properties and objects not affected by $a$, i.e., it is not influenced or informed by them."
> > > > >
> > > > > **"W4. Comparison to BISCUIT
> > > > > Can you provide more concrete reasons for not comparing? I am not insisting on comparing."**
> > > > >
> > > > > 1. In BISCUIT's model, each interaction variable is tied to a causal variable (e.g., an object). In contrast, for CDE, we learn a general action representation that not only applies to different objects, but can generalize to unseen ones.
> > > > >
> > > > > 2. In CDE, we try to learn action representations, while BISCUIT receives the actions as input. In more detail, in addition to the observed output, a "regime" variable R is required for BISCUIT, that corresponds to the action. Using the example of the concept figure from the paper of Lippe et al., the model receives as input a pair of images as well as the regime variable, which is "OpenObject". Given a dataset consisting of sequences of regime variables and images, the model learns to predict binary interaction variables $I_i$ (indicating that the regime variable affects object $i$) as well as a set of causal variables, $C_i$ (the objects and their states, in the case of iTHOR). In other words, as mentioned in Section 6.3 of Lippe et al., 2023, the goal of BISCUIT is to learn the causal variables (i.e., object and their states) given a sequence of known actions performed by an embodied agent to the objects of a scene. In contrast, the goal in CDE is to learn generalizable action representations given image pairs resulting from interactions with arbitrary objects.
> > > > >
> > > > > Given these two fundamental differences, it is not clear to us how we could directly set up a meaningful and fair comparison of our method with BISCUIT.
> > > > >
> > > > > **"W5. Questions on experiments (a-f)"**
> > > > >
> > > > > Regarding vanilla-R and vanilla-V:
> > > > >
> > > > > Vanilla models, drawn directly from the causal triplet paper, use concatenation of the image representations to perform action classification. In contrast, in Table 3, the CE only result uses the representation derived by CDE.
> > > > >
> > > > > Regarding the kNN experiment:
> > > > >
> > > > > The goal of the kNN experiment was to indicate that the CDE representations of the same action (implemented in different objects / conditions, including OOD) are close in the latent space. Based on previous remarks, this message was not clearly conveyed, and we decided to remove the experiment to avoid confusion.
> > > > >
> > > > > **"Q4. (c) This formulation of partial invertibility required for CDE must be exactly detailed under the assumptions."**
> > > > >
> > > > > Thank you for this comment this is now clearly stated in page 5 where we indicate that $z_a$ must be identifiable by the encoder.

---

### Official Review · Reviewer_RYXt · 2025-10-31

**Soundness:** 3
**Presentation:** 3
**Contribution:** 2
**Rating:** 6
**Confidence:** 3

**Summary:**

The paper introduces the concept of delta embeddings to represent atomic causal interventions between two images (e.g. open or close a drawer on the image) and proposes a simple method for classifying intervention actions. The proposed method first separately generates embeddings of a pair of images distant by a single intervention using a backbone Vit coupled with a MLP head tasked to disentangle the embeddings into delta embeddings. Then, the difference between the two delta embeddings is computed and sent as an input to an action classifier. The pipeline is trained end-to-end and regularized with contrastive and sparsity losses to improve the creation of robust representations that hold out-of-distribution. The method achieves improved o.o.d performance compared to baselines and generates embeddings with meaningful relationships between classes of actions.

**Strengths:**

1. The paper tackles a challenging problem in causal representation learning, namely the disentanglement of interventions, using a an original approach. The geometry of Delta embeddings could potentially convey very meaningful information, as hinted by the experiments and visualizations in the appendix.
2. The paper is well-written and easy to understand. The theoretical section complements well the description of the approach, justifying it accurately.
3. The experiments on out-of-distribution splits are particularly useful for assessing the generalization of the proposed delta embeddings.

**Weaknesses:**

Experiments are conducted on a single benchmark (causal Triplet), which limits the generalizability of the findings (although the experiments in o.o.d settings mitigate this issue). Using larger backbone models on more datasets would further strenghten the contributions.

**Questions:**

1. How realistic is the assumption of independence of latent factors in the data generative process? Indeed, the presence or absence of an object can have an effect on the lighting of the scene or on objects on top of it.
2. Must the set of possible actions $\mathcal{A}$ be known in advance or can the approach generalize to new actions, e.g. compositions of actions?
3. Have you conducted experiments with additional backbone models, e.g. ResNet-18, for a fair comparison with other baselines?
4. How do you interpret the drop in performance in o.o.d in Table 2 for the procTHOR multi-object setting?

---

> ### Author Response · Authors · 2025-11-19
>
> Thank you for carefully reading our paper, for your positive remarks and for your feedback for improvement. Following the review comments, we have uploaded a revised version of our paper which further clarifies the theoretical framework, provides experiments with additional backbones and explores the effect of actionable counterfactuals on our method. As such the updated version addresses several of the concerns raised. In more detail:
>
> **"Experiments are conducted on a single benchmark (causal Triplet), which limits the generalizability of the findings (although the experiments in o.o.d settings mitigate this issue). Using larger backbone models on more datasets would further strenghten the contributions."**
>
> Indeed the causal triplet is used as the most appropriate benchmark for our method. Please note that no directly applicable benchmark that includes a real-world dataset exists, to the best of our knowledge. Also, this benchmark includes three different datasets (two sets of data, one with two alternative settings), including a dataset derived from real-world videos (EpicKitchens), so the evaluation is performed on diverse data. Regarding the backbone model, we also added experiments using a CLIP backbone model and also another ViT pretrained with MAE. We have revised the experiments section to include results from these two backbone architectures.
>
> **"How realistic is the assumption of independence of latent factors in the data generative process? Indeed, the presence or absence of an object can have an effect on the lighting of the scene or on objects on top of it."**
>
> Our model operates under the assumption of the independent causal mechanisms (ICM, Scholkopf et al., 2021) which in our context translates into the assumption that there are different independent objects in a scene, which may also have a set of independent attributes. This is also an assumption made in previous work Liu et al., 2023. In the example mentioned regarding lighting, the model aims to learn a representation that is invariant to light, if the action is not affected by lighting. If the action is related to lighting (e.g., turn on light), then we assume that the object composition in the scene will not change before and after the lighting, so an invariant action representation could still be obtained. In any case, the exact modeling assumptions made in the proposed method have now been more clearly specified, while a theoretical investigation of the effect of noise has also been added (Sections 5.1.3 and Appendix E).
>
> **"Must the set of possible actions A be known in advance or can the approach generalize to new actions, e.g. compositions of actions?"**
> In the presented method, actions $A$ are known in advance and the CE loss enables predictions on the test dataset. The latent space could also be structured without known action labels using the contrastive loss only, and then used in a downstream task, although this may require the definition of an additional benchmarking framework with additional data to support unsupervised pretraining.  Exploring algebraic operations on actions in the latent space (e.g., compositionality, addition, subtraction etc) is a natural direction for future work, with potential applications in other ML domains and has been added in the conclusions (Section 7).
>
>
> **"Have you conducted experiments with additional backbone models, e.g. ResNet-18, for a fair comparison with other baselines?"**
> We have already conducted experiments with ResNet-18 and ViT backbone architectures. During this rebuttal period, we have also extended our experimental section with another ViT backbone architecture (pretrained with MAE) and also with a CLIP backbone. We have included the results in Section 6 of the updated paper. Results are similar and in some cases improved compared to the ViT-DINO backbone.

---

> > ### Author Response · Authors · 2025-11-19
> >
> > **"How do you interpret the drop in performance in o.o.d in Table 2 for the procTHOR multi-object setting?"**
> > Table 2 accounts for the multi-object setting where multiple objects are appeared in an image. This setting suggests two sub-problems: i) find the intervened object(only one object is intervened), out of all objects in the scene, ii) find the action that caused the intervention. Our PatchWiseModel achieves state of the art results and even beats the Oracle-Mask approach. The Oracle-Mask method provides a ground truth mask to the intervened object, so it provides almost full information for sub-problem (i). However our method can effectively isolate the intervened object and classify the action without ground truth mask supervision. Of course we observe that the performance declines between the two settings, but still is state of the art in this benchmark. Notice that the results have been updated (although they are similar) after a slight change that has been made in the method (now the loss is first applied to each patch and then the losses are aggregated, which is more meaningful theoretically).
> >
> > We hope that the revised version of the paper as well as the responses to the reviewer comment have addressed the reviewer's concerns.

---

> > > ### Comment · Reviewer_RYXt · 2025-11-25
> > >
> > > I thank the authors for their detailed responses to the comments and questions, all of which have been adequately answered. I will maintain my positive score.

---

### Official Review · Reviewer_Xc8e · 2025-11-04

**Soundness:** 4
**Presentation:** 3
**Contribution:** 4
**Rating:** 8
**Confidence:** 4

**Summary:**

This paper tackles the task of learning robust representations via interventional data. It proposes Causal Delta Embeddings (CDE) for representing interventions as latent-space deltas between pre/post-states. Results on synthetic and real benchmarks show gains in triplet evaluations and effective identification of changed semantics.

**Strengths:**

* tackles a very important problem of learning robust and interpretable representations
* leverages pretrained vision encoders in a good way and moves away from toy-dataset-only evaluations.
* clear conceptual framing of intervention representation problem
* strong quantitative OOD gains; well-executed ablations
* visualization & semantic structure analysis support claims

**Weaknesses:**

* only evaluates one vit backbone
* requires heavy supervision that is only possible with synthetic data
* empirical gains limited in real-world
* lacks exploration of confounding effects or imperfect interventions

**Questions:**

* how robust is CDE to noise or partial observability in interventions?
* does sparsity regularization risk collapsing subtle but real effects?
* how does the performance change when using different pretrained models? MAE, VQ-VAE's encoder, CLIP  would be very interesting

---

> ### Author Response · Authors · 2025-11-19
>
> Thank you for your comments/feedback and the careful assessment of our work. Thank you also for identifying the significance of this problem. Following the review comments we have performed additional experiments and more thorough theoretical investigation and have provided an updated version of the paper. Regarding the weaknesses mentioned:
>
> **"only evaluates one vit backbone"**
>
> Indeed, following this comment (and similar comments from other reviewers) we have added experiments with CLIP and MAE. Results are similar and in many cases improved compared to the ones attained using ViT-DINO. Please refer to the updated paper for result details.
>
> **"requires heavy supervision that is only possible with synthetic data"**
>
> Although controlled synthetic datasets are used, the EpicKitchens dataset is derived from real-world videos by taking frames before and after an action. Following this approach, it is relatively easy to apply delta embeddings to real data as well (as has been done in the paper). However, as the reviewer implies, the assumption of equal noise before and after the intervention does not hold in this case. These image pairs are referred to as "actionable counterfactuals" in Liu et al. 2023. We clearly distinguish these two cases in the updated version of the paper. Moreover, we have added a proof in Appendix E, that shows that if the exogenous noise is independent with zero mean, then the model trained with CE loss ignores the representation dimensions except $\tilde{z}_a - z_a$.
>
> **"empirical gains limited in real-world"**
>
> The method achieves a $+6\%$ with ViT-DINO and $+7\%$ with CLIP. Although these improvements are less impressive than the ones achieved in ProcTHOR, they are still substantial.
>
> **"lacks exploration of confounding effects or imperfect interventions"**
>
> Thank you for this comment. This was indeed missing in the paper. We have revised the text to clearly separate between perfect and actionable counterfactuals (resulting from imperfect interventions) and have also added Section 5.1.3 and Appendix E which explore how our method behaves in these cases. It is shown that under the assumptions of independent zero mean additive noise, the classifier trained using CE loss ignores the delta embedding dimensions that are not related to the action $a$.
>
> **"how robust is CDE to noise or partial observability in interventions?"**
>
> For noise, please see the response above. Regarding observability, the EpicKitchens dataset includes cases with object occlusion. The development of mechanisms for improving the method's robustness to occlusion scenarios has not been explored in this paper and is left as future work. We have added this note in the future work part of Section 7.
>
> **"does sparsity regularization risk collapsing subtle but real effects?"**
>
> Indeed, $L_1$ regularization does penalize all dimensions equally, which could suppress subtle causal signals. However, the ablation study shows that the sparsity regularization improves the OOD accuracy providing an argument supporting its use. More importantly, it is leads to representations that are consistent with sparse mechanism shift assumption, allowing further research under the causal representation learning framework.
>
> We believe that the modifications made to the paper, along with our replies, effectively resolve the issues raised by the reviewer.

---

### Author Response · Authors · 2025-12-01
**Overall Comment**

We would like to thank all reviewers for carefully reading our paper and providing constructive criticism, comments and suggestions. We have uploaded a revised version of the paper that includes additional experiments, theoretical results and updated text in response to these comments and believe that the manuscript is significantly improved as a result.

Note to the AC: Reviewer AZZ2 indicates in his/her original review that (s)he intends to raise the review score if his/her comments are answered. The comments were addressed and the reviewer raised the score from 4 to 6 (as also indicated in the followup response). However, this is not currently reflected due to the reverting of the scores by the ICLR organizers following the security incident. We hope that the AC will take this into consideration.

---

### Meta-Review · Area_Chair_Ca3o · 2026-01-06

**Summary:**

This paper proposes using causal delta embeddings to represent interventions for learning representations from actionable counterfactual pairs. Three reviewers gave positive scores, while one reviewer issued a weak reject and indicated a willingness to raise their score if the concerns were addressed. Reviewer AZZ2 primarily questioned the requirement for counterfactual pairs and requested greater clarity regarding the experimental settings. The authors clarified the experimental setup and explained the distinction between actionable counterfactuals and perfect counterfactuals. Overall, the paper makes a solid contribution to learning robust representations from counterfactual data.

**Reviewer Concerns:**

### Addressed Concerns

1. **Broader backbone evaluation:**
   The authors added experiments using multiple backbone architectures beyond ViT.

2. **Experimental clarity:**
   The authors clarified the datasets, baselines, and experimental settings, addressing concerns about reproducibility and interpretation.

3. **Ablation studies:**
   An ablation study was conducted to analyze the contributions of the three loss terms.

### Outstanding Concerns

1. **Requirement for heavy supervision (Reviewers AZZ2, Xc8e):**
   The method relies on heavily supervised counterfactual pairs, which may limit its applicability to real-world scenarios.

**Reviewer Scores:**

Reviewer **Xc8e** raised concerns about the reliance on a ViT backbone and the method’s applicability to real-world scenarios. In response, the authors added experiments using CLIP and MAE backbones. With these additions, the reviewer is likely to remain positive or potentially raise their score.

Reviewer **RYXt** questioned whether the core assumptions hold in real-world settings and requested analysis across more backbone architectures. The authors clarified that their assumptions follow prior work (e.g., Liu et al., 2023) and supplemented the paper with additional backbone experiments. This response is likely sufficient for the reviewer to remain positive.

Reviewer **AZZ2** expressed concerns about the reliance on counterfactual pair supervision and requested clarification of the experimental settings. The authors addressed these points by explaining the distinction between actionable and perfect counterfactual pairs and by clarifying the experimental setup. As a result, this reviewer may raise their score to positive.

Reviewer **jtzb** noted the lack of an ablation study on the three loss components. The authors conducted the requested ablation experiments, adequately addressing this concern. This reviewer is expected to remain positive.

---

### Decision · Program_Chairs · 2026-01-26

Accept (Poster)